

# Predicting maximal oxygen uptake from a 3-minute progressive knee-ups and step test

Yu-Chun Chung[1], Ching-Yu Huang[2], Huey-June Wu[3], Nai-Wen Kan[1], Chin-Shan Ho[4], Chi-Chang Huang[4] and Hung-Ting Chen[5]

[1] Center of General Education, Taipei Medical University, Taipei, Taiwan
[2] Service Systems Technology Center, Industrial Technology Research Institute, Hsinchu, Taiwan
[3] Department of Combat Sports and Chinese Martial Arts, Chinese Culture University, Taipei, Taiwan
[4] Graduate Institute of Sports Science, National Taiwan Sport University, Taoyuan, Taiwan
[5] Physical Education Office, Ming Chuan University, Taipei, Taiwan

## ABSTRACT

**Background**. Cardiorespiratory fitness assessment is crucial for diagnosing health risks and assessing interventions. Direct measurement of maximum oxygen uptake ($\dot{V}O_2$ max) yields more objective and accurate results, but it is practical only in a laboratory setting. We therefore investigated whether a 3-min progressive knee-up and step (3MPKS) test can be used to estimate peak oxygen uptake in these settings.

**Method**. The data of 166 healthy adult participants were analyzed. We conducted a $\dot{V}O_2$ max test and a subsequent 3MPKS exercise test, in a balanced order, a week later. In a multivariate regression model, sex; age; relative $\dot{V}O_2$ max; body mass index (BMI); body fat percentage (BF); resting heart rate (HR0); and heart rates at the beginning as well as at the first, second, third, and fourth minutes (denoted by HR0, HR1, HR2, HR3, and HR4, respectively) during a step test were used as predictors. Moreover, $R^2$ and standard error of estimate (SEE) were used to evaluate the accuracy of various body composition models in predicting $\dot{V}O_2$max.

**Results**. The predicted and actual $\dot{V}O_2$ max values were significantly correlated (BF% model: $R^2 = 0.624$, SEE = 4.982; BMI model: $R^2 = 0.567$, SEE = 5.153). The BF% model yielded more accurate predictions, and the model predictors were sex, age, BF%, HR0, $\Delta$HR3−HR0, and $\Delta$HR3−HR4.

**Conclusion**. In our study, involving Taiwanese adults, we constructed and verified a model to predict $\dot{V}O_2$ max, which indicates cardiorespiratory fitness. This model had the predictors sex, age, body composition, and heart rate changes during a step test. Our 3MPKS test has the potential to be widely used in epidemiological research to measure $\dot{V}O_2$ max and other health-related parameters.

## INTRODUCTION

In 2016, the American Heart Association launched a series of publications promoting the clinical evaluation of cardiorespiratory fitness (CRF) with the overall aim of improving the prevention and treatment of cardiovascular disease (CVD; *Ross et al., 2016*). Furthermore,

Corresponding author
Hung-Ting Chen,
simondr@mcu.mail.edu.tw

the association urged the US federal government to compile a registered CRF database (*Kaminsky et al., 2013*); this highlights the importance of CRF. CRF is generally defined as the integrated ability to transport oxygen from the atmosphere to the mitochondria for physical activity. Notably, CRF involves the respiratory, circulatory, and neuromuscular systems and has a clear and direct relationship with the functions of various systems. Individuals with weak CRF have an up to 70% all-cause mortality rate and 56% cardiovascular mortality rate (*Kodama et al., 2009*). Similarly, every 1-MET increase in athletic ability reduces all-cause mortality and cardiovascular mortality rates by 15% and 13%, respectively (*Kodama et al., 2009*). Numerous studies have suggested that CRF and CVD are related to all-cause mortality and cancer mortality (*Blair et al., 1989*; *Laukkanen et al., 2004*; *Sui, LaMonte & Blair, 2007*; *Sawada et al., 2014*; *Sui, LaMonte & Blair, 2007*). A recent meta-analysis reported CRF to be a predictor of the risk of sudden cardiac death (*Jiménez-Pavón, Lavie & Blair, 2019*). Therefore, CRF assessment is crucial for diagnosing health risks and assessing interventions.

CRF can be measured using the respiratory data of exercising participants. Specifically, these data are used to calculate maximal oxygen uptake ($\dot{V}O_2$ max), the gold standard for CRF measurement; in the measurement, participants either run on a treadmill or use an ergometer at an exercise intensity that increases progressively until a given maximum is reached. Although submaximal exercise models and nonexercise models (without an exercise test) are alternatives for estimating $\dot{V}O_2$ max in measuring CRF (*Abut, Akay & George, 2016*), the direct measurement of $\dot{V}O_2$ max yields more objective and accurate results. However, such measurement is inconvenient because it requires expensive equipment and well-trained experimenters. In addition, participants perceive such measurement tests to be exhausting, time-consuming, and relatively risky and are thus less willing to participate. Accordingly, researchers have developed various submaximal exercise tests to indirectly estimate $\dot{V}O_2$ max; moreover, retrospective studies conducted by the American Heart Association have demonstrated that CRF indicators, whether directly measured or indirectly estimated, are robust indicators of health (*Ross et al., 2016*).

Submaximal exercise is a common method for estimating $\dot{V}O_2$ max, particularly in epidemiological research and large-scale physical fitness testing that involve numerous participants. The field tests in these measurement procedures include running, shuttle running, and the step test, with the step test being the most common method for evaluating cardiovascular function (*Grant, Joseph & Campagna, 1999*). In particular, the YMCA step test is widely used to predict $\dot{V}O_2$ max (*Beutner et al., 2015*). Currently, the Sports Administration of Taiwan's Ministry of Education uses the 3-min Harvard step test for its National Physical Fitness and Cardiovascular Test. Specifically, three heart rate measurements are used to calculate the step-up index. However, previous studies have reported considerable differences in the validity of using the step test index to evaluate $\dot{V}O_2$ max, with the corresponding correlation coefficient (R) being 0.35–0.94 (*Buckley et al., 2004*; *Chang & Lin, 1995*; *Mazic et al., 2001*; *Su, Lin & Hsieh, 2006*; *Chang & Lin, 1995*; *Yoopat, Vanwonterghem & Louhevaara, 2002*). Furthermore, step tests require the use of step-up boxes, and the overall test time must be at least 6 min to allow for heart rate recovery. Participants who are less physically fit or who have knee conditions may find it

difficult to complete the test and may also fall in the process of going up and down the stairs. A team of Japanese researchers developed a new 3-min walking test (*Cao et al., 2013*). Specifically, their main evaluation criteria comprised participant characteristics such as age, sex, and BMI as well as participants' RPE during exercise. These criteria were determined to be effective predictors of $\dot{V}O_2$ max, and participants thought that this method was quicker and easier.

Tests of general CRF are crucial to the clinical evaluation of CVD. Additionally, the advantages and disadvantages, such as venue size, participant willingness, and the instruments, of various past field tests should be considered during the formulation of new methods, as done in the present study. Accordingly, we conducted the present study with the aim of developing a rapid, convenient, and low-risk model that can predict $\dot{V}O_2$ max in Taiwanese adults. Additionally, our model accords with the principle that physical exercise ought to be progressive. We investigated the feasibility of using a 3-min progressive knee-ups and step (3MPKS) test to predict $\dot{V}O_2$ max.

## MATERIALS AND METHODS

### Participants

Prospective participants were excluded if they (1) had cardiovascular, pulmonary, or metabolic diseases; (2) had neurological, muscular, or skeletal disorders that affected their athletic ability; (3) had other health conditions that made them unsuited for moderate or intense exercise; or (4) were taking medications that could affect the outcome of this study. In total, among 200 participants recruited for this experiment, 166 completed the test. The data of the 166 participants were included in the analysis (age: 20–64 years; 65 men, 101 women). Among the 34 participants excluded, one participant withdrew from the experiment after experiencing suspected symptoms of arrhythmia during exercise; 11 were excluded because they failed to complete the step test within the requisite time (3 min); 12 were excluded because they could not attain the requisite step frequency and knee height for 20 consecutive seconds; nine were excluded because they had missing or improperly measured heart rate data; and one was excluded for having a ''0'' in their heart rate data. All participants signed an informed consent form after understanding their rights, the risks when participating in this study, and the purpose and method of our research. Our research plan was approved by the Institutional Review Boards (IRBs) of the Industrial Technology Research Institute and of Taipei Medical University (IRB No: N201808055). Participant characteristics are detailed in Table 1.

### Procedure

The anthropometric and body composition measures were height, weight, and body fat percentage (BF%). BF% was measured using bioelectrical impedance analysis (InBody 720, Biospace, USA; *McLester et al., 2020*), and body mass index (BMI, in kg/m$^2$) was calculated as the quotient that is weight (in kilograms) divided by the squared height (in meters).

We conducted two exercise tests in a counterbalanced design. The second test was conducted exactly 1 week after the first and at the same time of the day to ensure that the participants recovered adequately from the first exercise. The participants underwent 5–10

**Table 1  Participant characteristics.**

|  | Total | Training dataset | Testing dataset |
|---|---|---|---|
| Sample size(n) | 166 | 124 | 42 |
| Age(years) | 41.9 ± 9.6 | 42.2 ± 9.4 | 40.8 ± 10.2 |
| Male (n) | 65 | 44 | 21 |
| Height | 164.83 ± 8.35 | 164.33 ± 8.07 | 166.30 ± 9.06 |
| Weight | 65.63 ± 13.60 | 65.22 ± 14.08 | 66.85 ± 12.15 |
| Body fat (%) | 27.81 ± 7.92 | 27.79 ± 7.65 | 27.86 ± 8.75 |
| $\dot{V}O_2$ max (ml kg$^{-1}$ min$^{-1}$) | 34.45 ± 8.69 | 34.06 ± 8.14 | 35.61 ± 10.15 |
| HR0 | 86.04 ± 12.78 | 86.04 ± 12.99 | 86.02 ± 12.29 |
| ΔHR3- HR0 | 71.00 ± 13.24 | 71.10 ± 13.41 | 70.69 ± 12.87 |
| ΔHR3-HR4 | 14.64 ± 13.72 | 14.14 ± 13.94 | 16.65 ± 14.09 |

**Notes.**

Data are presented as mean ± standard deviation.

HR0,  heart rate at the beginning; ΔHR3-HR0, difference between third minute heart rate and beginning heart rate; ΔHR3-HR4, difference between third minute and fourth minute heart rates.

min of dynamic warm-up prior to both exercise tests; to mitigate extraneous influence on the results, the participants were also asked not to engage in moderate or intense exercise 48 h before both exercise tests.

To measure the $\dot{V}O_2$ max of the participants, we used a bicycle ergometer (839E, Monark, Varberg, Sweden) for a maximal graded exercise test. After participants sat still for 2 min, they sat on the stationary bicycle and started cycling at the speed of 70 ± 10 rpm. The participants began the exercise with a 2-min warm-up at 25 W loading, where the loading was increased by 15 W every 2 min. The testing was terminated when the participants could no longer continue the exercise due to bradypnea or fatigue, although the bicycle speed was maintained at 70 rpm. Subsequently, the participants rested for 3 min at a loading of 0 W (no resistance). Throughout the exercise testing, the participants wore a watch to monitor their heart rate and a mask to monitor their breathing. Breath-by-breath analysis was conducted on the participant data through a cardiopulmonary testing system (MetaMax 3B, Cortex, Germany). $\dot{V}O_2$ max was defined as the maximum average oxygen uptake for 20 consecutive seconds. To ensure that every participant reached $\dot{V}O_2$ max, we defined $\dot{V}O_2$ max as being reached if two of the three following conditions were met: (1) $\dot{V}O_2$ plateaus with increases in work rate; (2) the maximum respiratory exchange ratio is ≥1.10; and (3) 90% of the expected maximal heart rate, obtained by subtracting the participant's age from 220, is reached (*American College of Sports Medicine, 2009*). Nearly all participants satisfied the criteria for an acceptable $\dot{V}O_2$ max, with only one participant excluded from the $\dot{V}O_2$ max test due to suspected symptoms of arrhythmia observed in the step test.

### 3MPKS test

Prior to the 3MPKS test, the participants wore a sports watch with heart rate (Polar V800, USA) and stride sensors (Polar S3 BlueTooth Stride Sensor, USA). The heart rate sensor was placed at the center of each participant's chest using a heart rate belt (Polar H10), and the step sensor was fixed on a pair of shoes, with shoelaces, to monitor their heartbeat

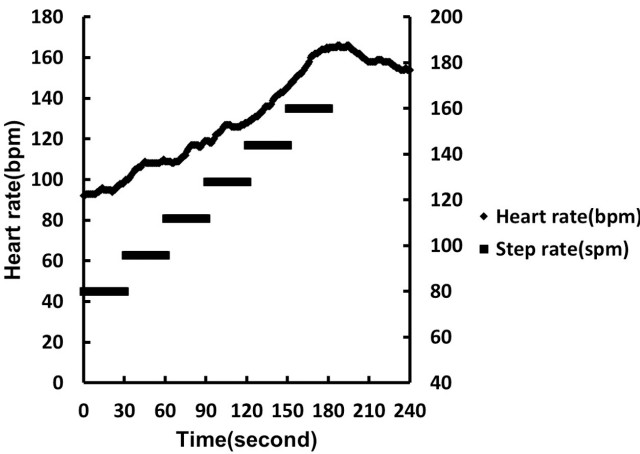

**Figure 1  3MPKS heart rate model and corresponding step frequency.**

and number of steps taken. After the devices were worn, we measured the midpoint of the line connecting the anterior epicondyle to the midpoint of the sacrum. We marked the midpoint on the wall using colored tape as a reference for the height at which the knee should be lifted to when stepping. After the test started, the participants followed the appropriate rhythm and were required to lift their knee to the marked height at each step. The participants began the test at a pace of 80 spm (steps per minute), which increased by 16 spm every 30 s in six stages. The participants walked in stages 1 to 4 and had to perform stationary running in stages 5 and 6 (Fig. 1). We stopped the exercise if the participants could not achieve the requisite knee height or rhythm for 30 s. For their safety, the participants were asked to relax at a step rate of 80 spm in the first 30 s before resting in a standing position. We recorded the participants' heart rate during the exercise, at the end of the exercise, and 1 min after the end of the exercise. Thirty-four participants were excluded because (1) their heart rate data were missing, (2) their heart rate was 0, (3) they did not maintain the requisite step frequency or knee height for 20 consecutive seconds, (4) they failed to complete the step test within the requisite duration, and (5) they were suspected of having heart arrhythmia. Potential predictor variables for the results of the 3MPKS test were based on per-second heart rate data collected during the test. The data included heart rate at the beginning as well as at the first, second, third, and fourth minutes, denoted by HR0, HR1, HR2, HR3, and HR4, respectively, and were used for subsequent analysis.

## Statistical analyses

To construct and subsequently evaluate a model for estimating relative oxygen uptake, we divided the full sample set ($n = 166$) into a 75% training sample set ($n = 124$) and 25% test sample set through simple random sampling. We analyzed the descriptive statistics for the main parameters, for the whole sample, and for the two subsamples.
## Development of prediction model

Using Pearson correlation coefficients, we examined the relationship between the predicted and actual relative oxygen uptakes. Multiple regression analysis was used to construct a method for selecting which variables to include in the model for predicting relative oxygen uptake. Through a backward-selection regression approach, the initial model included all possible predictors, including sex (men = 1, women = 0), age, BMI, BF%, HR0, HR1, HR2, HR3, HR4, $\triangle$HR0 − HR1, $\triangle$HR1 − HR2, $\triangle$HR2 − HR3, $\triangle$HR3 − HR0, and $\triangle$HR3 − HR4. Additionally, we constructed a BMI model and BF% model to predict body composition. The goodness of fit and precision of the regression equations were evaluated using the multiple coefficient of determination ($R^2$), absolute standard error of estimate (SEE), and relative SEE (%SEE).

To construct an accurate regression model, the regression assumptions were verified. We conducted a Kolmogorov–Smirnov test to examine the normality of the residuals, and we calculated the variation inflation factor (VIF) to check for multicollinearity.

All statistical analyses were performed using SPSS version 20 (IBM, USA). Statistical significance was indicated by an alpha level of 0.05.

## RESULTS

The 166 participants had an average age of 41.9 ±9.6 years (range: 22–64 years), and 40% of them were men. Their mean relative oxygen uptake was 34.45 ±8.69 mL/kg/min. The training sample and test sample did not differ significantly with respect to their parameter values ($p > 0.05$) Table 1.

The test–retest reliability of the 3MPKS test, as evaluated in our laboratory, was excellent: the intraclass correlation coefficient (ICC) was 0.88 (95% confidence interval [CI]: 0.77–0.94), and 60 Taiwanese adults tested 1 week apart participated in this evaluation. In general, good, moderate, and poor reliability levels are indicated by ICC values of >0.75, 0.5–0.75, and <0.5, respectively.

According to the correlation matrix, $\dot{V}O_2$ max had the strongest correlation with BF% among all variables ($R = -0.662$; training data set, $n = 124$). In addition, $\dot{V}O_2$ max was significantly correlated with the heart rate parameters (HR0, HR2, HR3, and HR4), whose data were collected in the step test. $\dot{V}O_2$ max was most and least correlated with HR4 ($R = -0.442$) and HR3 ($R = -0.289$), respectively. Despite the high correlation between $\dot{V}O_2$ max and the heart rate parameters at different stages, the heart rates of the participants were expected to increase continuously from the first to third minutes of stepping, if performed properly. An individual's heart rate typically reaches its peak immediately after exercise, and it either decreases at 1 min after exercise or does not decrease at all depending on whether the individual recovers quickly or poorly. Because heart rate is dynamic, to establish a regression model, we used combinations of heart rate parameters and adopted the difference between predicted and measured heart rate data at each stage as inputs (Table 2).

The results of our other cross-validation analyses are presented in terms of CE (Constant error) values. The absolute CE values for subgroups stratified by sex and age were <1.00

**Table 2 Correlation between $\dot{V}O_2$ max and features in training dataset ($n = 124$).**

| | $\dot{V}O_2$ max | Sex | Age | BMI | BF% | HR0 | HR1 | HR2 | HR3 |
|---|---|---|---|---|---|---|---|---|---|
| Sex | 0.597** | | | | | | | | |
| Age | −0.342** | −0.114 | | | | | | | |
| BMI | −0.083 | 0.334** | −0.160 | | | | | | |
| BF% | −0.662** | −0.491** | 0.109 | 0.448 | | | | | |
| HR0 | −0.317** | −0.242* | −0.101 | −0.058 | 0.227* | | | | |
| HR1 | −0.344** | −0.033 | −0.283* | −0.039 | 0.274* | 0.69** | | | |
| HR2 | −0.357** | −0.312** | −0.093 | −0.005 | 0.308** | 0.592** | 0.899** | | |
| HR3 | −0.289* | −0.254* | −0.21* | 0 | 0.248* | 0.525** | 0.725** | 0.8** | |
| HR4 | −0.442** | −0.42** | −0.13 | −0.063 | 0.334** | 0.564** | 0.57** | 0.629** | 0.702** |

Notes.

BF%, body fat percentage.

**Correlation coefficient is significant($p < 0.001$).

*Correlation coefficient is significant($p < 0.05$).

for the two models (both in training and testing data sets, $n = 124$ and 42). Regarding the subgroups stratified by $\dot{V}O_2$ max, the CE values were negative in low-fitness, middle-fitness subgroups in training data set and low-fitness in testing data set. On the other hand, the CE values were positive in high-fitness in all two data sets (Table 3).

Figures 2 and 3 present the Bland–Altman plots produced by the BF% and BMI models based on the testing data set ($n = 42$). As evident in the plots, the differences between the predicted and measured data were within an acceptable range. The mean error of the BF% model was −0.36 mL/kg/min (95% CI [−12.38–11.98]). For the BMI model, the mean error was 0.4 mL/kg/min (95% CI [−12.35–13.58]). In the BF% and BMI models, the errors for three and two participants, respectively, fell outside the 95% CI.

We constructed a model to predict relative oxygen uptake by using multiple regression analysis. The parameters selected for the BF% model were sex, age, BF%, HR0, $\Delta$HR3 − HR0, and $\Delta$HR3 − HR4; $R^2 = 0.624$ and SEE = 4.982 (training data set, $n = 124$) (Fig. 4). The parameters selected for the BMI model were sex, age, BMI, initial heart rate, $\Delta$HR3 − HR0, and $\Delta$HR3 − HR4; $R^2 = 0.567$ and SEE = 5.153 (training data set, $n = 124$) (Fig. 5). We used BF% as a predictor of body composition; it is more accurate relative to BMI, which is calculated using only height and weight (Table 4). Table 4 presents the cross-validation results for the predicted residual sum of squares (PRESS) statistics ($R^2 p = 0.64$ and SEE $p = 4.84$), which demonstrated minimal shrinkage in the accuracy of the regression model.

All regression assumptions were satisfied in our $\dot{V}O_2$ max prediction models. Specifically, the Kolmogorov–Smirnov test indicated normality in the residuals ($p > 0.05$). No pattern was determined in the scatter plot between the residuals and predicted $\dot{V}O_2$ max. Multicollinearity was absent among the predictor variables: the VIF ranges for the BF% and BMI models were 1.09–1.49 and 1.10–1.40, respectively; multicollinearity is absent if VIF $\leq 10$ (O'brien, 2007).

**Table 3** Measured versus predicted $\dot{V}O_2$max constant error (CE) and standard deviations (SD) for subgroups of the training dataset and testing dataset.

| Subgroup | n(%) | BF% model(%) | | BMI model(kg m$^{-2}$) | |
|---|---|---|---|---|---|
| | | CE | SD | CE | SD |
| Training set(n = 124) | | | | | |
| Sex | | | | | |
| Female | 80(64.5) | −0.01 | 3.95 | 0.01 | 4.45 |
| Male | 44(35.5) | −0.02 | 6.23 | 0.01 | 5.99 |
| Age | | | | | |
| <40 years | 48(38.7) | −0.34 | 4.72 | −0.49 | 4.94 |
| 40–50 years | 44(35.5) | 0.21 | 5.35 | 0.22 | 5.24 |
| ≥50 years | 32(25.8) | 0.17 | 4.46 | 0.47 | 4.95 |
| $\dot{V}O_2$max | | | | | |
| <29 ml/kg/min | 34(27.4) | −2.77 | 3.19 | −3.13 | 3.66 |
| 29–38 ml/kg/min | 56(45.2) | −0.22 | 4.46 | −0.22 | 4.60 |
| ≥38 ml/kg/min | 34(27.4) | 3.09 | 5.18 | 3.51 | 4.75 |
| Testing set(n = 42) | | | | | |
| Sex | | | | | |
| Female | 21(50) | −0.15 | 5.85 | −0.89 | 5.84 |
| Male | 21(50) | 0.87 | 6.82 | 0.08 | 7.62 |
| Age | | | | | |
| <43 years | 24(57.1) | 0.67 | 6.22 | −0.07 | 6.8 |
| ≥43 years | 18(42.9) | −0.05 | 6.55 | −0.85 | 6.79 |
| $\dot{V}O_2$max | | | | | |
| <35 ml/kg/min | 22(52.4) | −2.88 | 4.95 | −3.86 | 5.31 |
| ≥35 ml/kg/min | 20(47.6) | 3.93 | 5.73 | 3.39 | 6.11 |

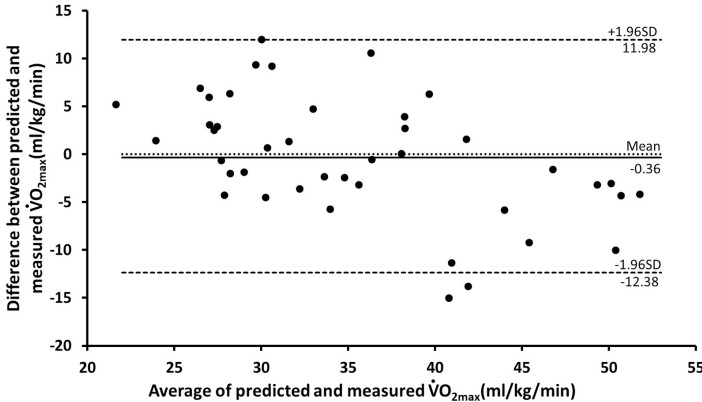

**Figure 2** Bland Altman plot, including limits of agreement, for predicted and measured $\dot{V}O_2$ max (ml/kg/min) of BF% model by testing dataset (n = 42). Black line mean difference. Dashed line±1.96 ×SD.

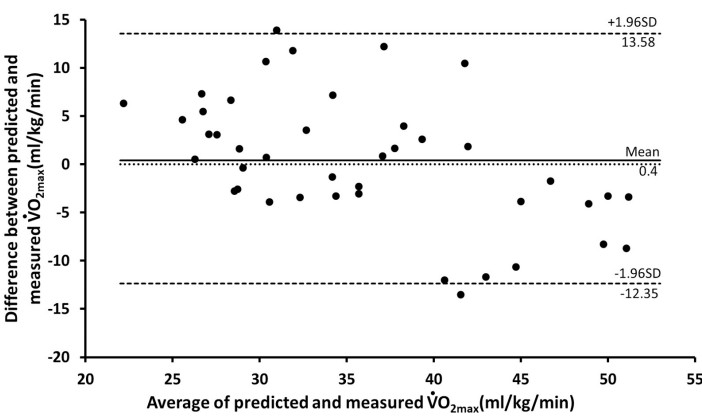

**Figure 3** Bland Altman plot, including limits of agreement, for predicted and measured V̇O₂ max (ml/kg/min) of BMI model by testing dataset (*n* = 42). Black line mean difference. Dashed line±1.96 ×SD.

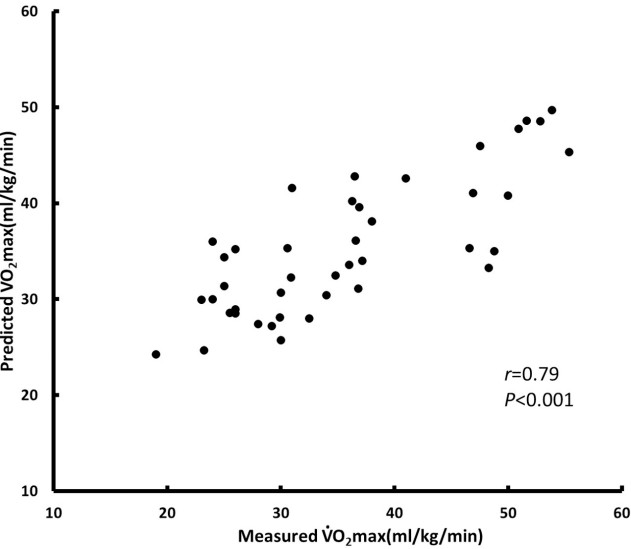

**Figure 4** BF% model for testing test (*n* = 42).

## DISCUSSION

This study developed a practical and easy-to-use model for predicting V̇O₂ max in Taiwanese people. We recruited 166 Taiwanese adults and constructed and then evaluated a prediction model. Our results suggest that age, sex, and BF% as well as heart rate during the step test are excellent predictors of V̇O₂ max. We also developed a novel 3MPKS test.

*Nes et al. (2011)* conducted large-scale V̇O₂ max tests on 4,260 participants. They developed a nonexercise model and determined four variables (age, waist circumference, physical activity, and resting heart rate) to be excellent predictors of V̇O₂ max; for their model, $R^2$ was 0.61 and SEE was 5.70 mL/kg/min for men, and $R^2$ was 0.56 and SEE was 5.14

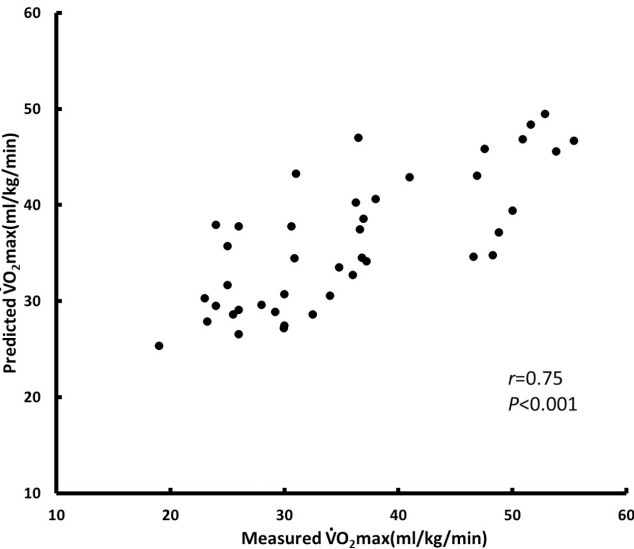

**Figure 5** BMI model for testing ($n = 42$).

**Table 4** Estimation of $\dot{V}O_2$ max through multiple regression model ($n = 124$).

| $\dot{V}O_2$max ($ml\,kg^{-1}\,min^{-1}$) | BF% model (%) | | | BMI model ($kg\,m^{-2}$) | | |
|---|---|---|---|---|---|---|
| | Coefficients | $\beta$ | $p$ value | Coefficients | $\beta$ | $p$ value |
| Constant | 72.334 | | .000 | 82.387 | | .000 |
| Sex (0=women, 1=men) | 4.366 | 0.258 | .000 | 9.338 | 0.551 | .000 |
| Age(yr) | −0.261 | −0.302 | .000 | −0.327 | −0.378 | .000 |
| Body composition | −0.448 | −0.421 | .000 | −0.718 | −0.346 | .000 |
| HR0 | −0.134 | −0.214 | .001 | −0.171 | −0.273 | .001 |
| ΔHR3- HR0 | −0.082 | −0.136 | .041 | −0.099 | −0.163 | .017 |
| ΔHR3-HR4 | 0.073 | 0.124 | .048 | 0.081 | 0.139 | .032 |
| $R^2$ | 0.624 | | | 0.567 | | |
| SEE | 4.982 | | | 5.153 | | |
| SEE% | 14.46 | | | 14.96 | | |
| PRESS | 2904.186 | | | 3107.325 | | |
| SEEp | 4.840 | | | 5.006 | | |
| $R^2p$ | 0.644 | | | 0.619 | | |

**Notes.**
BMI, body mass index; BF%, body fat percentage; $\beta$, standardized regression weights; SEE, standard error of estimate; SEE%, SEE / mean of measured $\dot{V}O_2$ max ×100.; PRESS, predicted residual error sum of squares; SEEp, PRESS standard error of estimate; $R^2p$, PRESS squared multiple correlation coefficient.

mL/kg/min for women. *Jackson et al. (2012)* conducted a 27-year study that examined the $\dot{V}O_2$ max of 11,365 people and used variables such as age, sex, BMI, waist circumference, resting heart rate, physical activity, and smoking habits to estimate CRF; for their model, $R$ was 0.78–0.81 and SEE was 5.3–5.6 mL/kg/min. Although the nonexercise model is

an excellent predictor of $\dot{V}O_2$ max, its SEE is generally higher than those of submaximal exercise models; compared with nonexercise models, our developed BF% model had better predictive performance and a lower standard error of estimate ($R^2 = 0.624$ and SEE = 4.982). *Abut, Akay & George (2016)* reported that (1) when perceived functional ability (PFA) was used as the sole predictor of $\dot{V}O_2$ max, an $R$ value of 0.73 and a higher RMSE of 6.08 mL/kg/min could be obtained; (2) when submaximal ending speed (SM-ES) of a treadmill was used as the sole predictor, the $R$ value increased to 0.82 and the RMSE was relatively low at 4.99 mL/kg/min; and (3) when both PFA and SM-ES were used as predictors, the $R$ value was 0.89 and RMSE was 4.14 mL/kg/min. These findings indicate that predicted values of $\dot{V}O_2$ max that are based only on participant self-reports are likely to deviate from their measured values. Although predictive performance is ostensibly improved when motion is added to the prediction model, the cost of exercise tests due to the use of this method restricts its application in large-scale tests.

Several studies have developed simple models involving submaximal motion. *Lee et al. (2019)* investigated 568 adults and used sex, age, height, and weight and inverse recovery heart rate during a YMCA step test to predict $\dot{V}O_2$ max; for their model, $R$ was 0.78 and SEE was 4.74 mL/kg/min. The duration of their exercise test plus recovery time was only 4 min, and they used exercise-induced heart rate as a predictor; their results are similar to ours. Their study provided a simple and practical method for simultaneously estimating CRF in many Korean adults. *Cao et al. (2013)* used age, sex, and physical composition as well as stepping distance over a 3-min period to develop a set of prediction methods. They determined that BF% (a measure of body composition) was a better predictor than BMI ($R^2 = 0.83$ vs. 0.80, SEE = 4.565 vs. 5.037 mL/kg/min). In contrast to our method, their method has the considerable advantages of a shorter testing time of 3 min and the fact that participants need not wear a heart rate monitor. However, their test is limited by its need for a 20-m open space. Similarly, we found that sex, age, and BF% as well as heart rate during the 3MPKS test yielded the best prediction performance ($R = 0.79$, SEE = 4.982 mL/kg/min). Because BMI is based on only height and weight and may not accurately represent the body characteristics of participants, BMI is a less accurate predictor than BF%.

Most submaximal exercise models proposed by previous studies involve a fixed-height step test. However, the height and leg length of participants when standing may affect their physiological response in the step test (*Culpepper & Francis, 1987*). Relative to their European counterparts, Asian adults have shorter heights and leg lengths when standing (*Stanfield et al., 2012*). Therefore, differences in heart rate and oxygen consumption potentially affect the model's prediction. The 3MPKS test employs the knee-ups and step test to measure the physical fitness and cardiopulmonary endurance of older adults (*Rikli & Jones, 2001*). In the test, participants must execute tasks at various knee heights based on their thigh length, and individualized exercise testing goals are provided. Moreover, most field tests involve average speed tests, such as step tests and running. In running tests specifically, if the distance is used as the capacity index but the speed or frequency of exercise is not progressively increased, participants may exercise intensely at the beginning of the test (i.e., run at a higher speed). However, due to the lack of appropriate speed

allocation, decremental loading occurs in participants as their physical strength decreases. The difficulty of diagnosing potential heart diseases in advance increases the risk of sudden death during running tests. To the best of our knowledge, research has not been conducted on the ethics of running tests. Most previous studies have investigated the rate of sudden death among athletes in long-distance competitions. However, cases of sudden cardiac death occur frequently worldwide during running tests, and the principle that physical activities ought to be progressive must be adhered to in physical fitness tests. Our research method used body composition and heart rate as variables. The advantages of the 3MPKS test are that it does not require a step-up box and is not subject to venue restrictions. These make the 3MPKS test accord with the principle that physical activities ought to be progressive, thus making it safer.

Considering the immediacy of heart rate measurement and that of confounding factors, we used a chest-worn heart rate monitor in the experiment. Although the requirement of heart rate monitoring constitutes a disadvantage for the 3MPKS test, it is ameliorated by the prevalence of low-cost wearable devices. More comfortable than the chest-worn heart rate belt, products that combine running clothes with heart rate belts have also appeared on the market. Research has also suggested a high correlation between the heart rate measurements of various types of optical devices and chest-worn heart rate belts (*Stahl et al., 2016*). Therefore, when conducting a large-scale cardiorespiratory general test, the use of easily wearable optical heart rate monitors can be considered. The whole-range monitoring of heart rate can also considerably improve test safety in a field study. Notably, through whole-range monitoring, we found that one research participant was likely to have an unknown heart disease. We then terminated the experiment for the participant and recommended that the participant seek medical treatment. This example illustrates a side benefit of CRF tests.

In our research model, heart rate during stepping at each stage was used as the main variable. Therefore, the test may be unsuitable for individuals who have psychological sensitivity or dysautonomia or who are taking medication. Furthermore, because our participants were adults between 20 and 64 years old, it was unclear whether our 3MKPS test is appropriate as a physical fitness and cardiorespiratory test for students (7–23 years old) and older adults (≥65 years old). Future research must include samples with greater diversity in age and ethnicity to assess whether our 3MKPS test can be applied to the wider global population.

## CONCLUSION

This study, involving Taiwanese adults, constructed and verified a model for predicting $\dot{V}O_2$ max, which is used to measure CRF. This model comprises the predictors sex, age, and body composition as well as heart rate changes during a step test. Our 3MKPS test has three advantages: it has a short testing time of 4 min, it has no venue limitations, and it does not require a step box. Furthermore, measurements can be taken for many participants simultaneously by asking them to wear a heart rate monitor and move according to a beat. Our model can also be applied to large-scale epidemiological research. In future

applications, the model can be combined with smartwatches or used to develop health and well-being apps, helping users to track their $\dot{V}O_2$ max. Future research can further explore the correlation between various diseases and $\dot{V}O_2$ max, as predicted using our simple and reliable method for measuring CRF.

### Funding
This research was supported by Research Grants from Taipei Medical University (no. TMU105-AE1-B06) and the Sports Administration, Ministry of Education, R.O.C. for the Comprehensive Research for the Industrial Technology Research Institute's Technology Fitness Program (no. J4653H1A20). The funders had no role in study design, data collection and analysis, decision to publish, or preparation of the manuscript.

### Grant Disclosures
The following grant information was disclosed by the authors:
Research Grants from Taipei Medical University: no. TMU105-AE1-B06.
Sports Administration, Ministry of Education, R.O.C. for the Comprehensive Research for the Industrial Technology Research Institute's Technology Fitness Program: no. J4653H1A20.

### Competing Interests
The authors declare there are no competing interests.

### Author Contributions
- Yu-Chun Chung conceived and designed the experiments, performed the experiments, analyzed the data, prepared figures and/or tables, authored or reviewed drafts of the paper, and approved the final draft.
- Ching-Yu Huang conceived and designed the experiments, analyzed the data, authored or reviewed drafts of the paper, and approved the final draft.
- Huey-June Wu conceived and designed the experiments, performed the experiments, authored or reviewed drafts of the paper, and approved the final draft.
- Nai-Wen Kan and Chi-Chang Huang performed the experiments, authored or reviewed drafts of the paper, and approved the final draft.
- Chin-Shan Ho performed the experiments, analyzed the data, authored or reviewed drafts of the paper, and approved the final draft.
- Hung-Ting Chen conceived and designed the experiments, analyzed the data, prepared figures and/or tables, authored or reviewed drafts of the paper, and approved the final draft.

### Human Ethics
The following information was supplied relating to ethical approvals (i.e., approving body and any reference numbers):
Institutional Review Boards (IRBs) of the Industrial Technology Research Institute and of Taipei Medical University (IRB No: N201808055).

## Data Availability

Raw measurements are available in the Supplemental Files.

## Supplemental Information

Supplemental information for this article can be found online at http://dx.doi.org/10.7717/peerj.10831#supplemental-information.

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
