# Peer review of "Predicting maximal oxygen uptake from a 3-minute progressive knee-ups and step test"

_PeerJ, doi:10.7717/peerj.10831_

## Round 0.1 · original submission · Major Revisions

As outlined by the reviewers below, several sections in your introduction require referencing to support claims made. In addition to addressing the reviewer concerns outlined below, specifically concerning details of your methodology and greater validation of your model, please also state:
- how the study population was recruited (e.g. how many were approached and declined or met exclusion criteria, where participants were recruited from);
- amount of missing data in your data set concerning heart rate variables and how this data was treated;
- how many individuals were not able to undertake the 3MPKS because knee height could not be achieved etc.

As outlined by Reviewer 1, please also reorganise your discussion to increase the focus on the results of your study.

Reviewer 1 ·

Basic reporting

Q. Clear and unambiguous, professional English used throughout
A. No. This work requires improvement in English

Q.Literature references, sufficient field background/context provided.
A. Yes

Q. Professional article structure, figures, tables. Raw data shared
A.Yes

Q.Self-contained with relevant results to hypotheses.
A. Yes

Experimental design

Q. Original primary research within Aims and Scope of the journal.
A.Yes

Q.Research question well defined, relevant & meaningful. It is stated how research fills an identified knowledge gap.
A. Yes

Q.Rigorous investigation performed to a high technical & ethical standard.
A. Yes

Q.Methods described with sufficient detail & information to replicate.
A. Yes

Validity of the findings

Q. Impact and novelty not assessed. Negative/inconclusive results accepted. Meaningful replication encouraged where rationale & benefit to literature is clearly stated.
A.

Q.All underlying data have been provided; they are robust, statistically sound, & controlled.
A.

Q.Conclusions are well stated, linked to original research question & limited to supporting results.
A.

Additional comments

This article requires improvement in English. Also it is strongly recommended to reorganize the article. Also, validation of the model should be performed in much more detail.

Abstract
- Line 26: I believe you excluded "BMI" for the second model.

Introduction
- Needs to be reorganized.
- Paragraph 2 should be rewritten. Before mentioning sub-maximal methods, you should clearly state the current gold standard for measuring maximal oxygen uptake.

Materials & Methods
- Line 96: You did not recruit 166 participants. I believe that is the number of participants after the exclusion. Please state the number of participants you have recruited.
- Line 128~130: Please state the reference for this exclusion criteria. Also, please mention how many participants were excluded according to this exclusion criteria.
- Line 147~150: Again, how many samples were excluded according to this criteria?
-Line 151: This should be in the result section.
Line 160~162: You should apply the predicted residual error sum of squares (PRESS) statistic for cross-validation method.

The model and features should be analyzed in more detail. Please provide following.
- Correlation matrix between features and VO2 max.
- Constant error (CE) and standard deviation for subgroups (i.e. Female, male, old, young, high VO2 max, low VO2 max, etc.)
- Bland-Altman plot

Discussion
- Discussion should be more about your study.
- Please reorganized the section for better readability.

Reviewer 2 ·

Basic reporting

The structure of the article is good. The tables and figures are relevant.

Overall the article is well written, but some sentences are unclear.

Some claims are not supported by adequate references

see specific remarks in the general comments

Experimental design

The research question is well defined and the experimental design is meaningful. Sufficient information is provided to replicate the study

The statistical analysis is relevant. I particularly appreciated the use of train and test sets to validate the model.

Validity of the findings

Given the large sample size and the valid statistical analysis, the findings seem valid and solid. Conclusions are supported by the results

Additional comments

General remark: VO2 should be written with a dot on the V to indicate that the value is a flux and not a volume. Ask for the editor for a correct display in the final version.

Abstract: HR0 HR3 and HR4 not defined. The word limit of PeerJ is 500. Please give a better introduction to show background information.

L 46 please add reference to support your claim
L 53 sentence unclear CRF cannot “measure” something
L 55 Not clear what subalgorithms and nonmotion algorithm are (definition or references)
L 61-63 Not sure to understand the meaning of the sentence. Do you mean that both direct measurements and indirect estimations are useful health outcome predictors? please clarify
L 66-67 Please add references
L 72 What is R? Please define.
L 106-107. Do you have a reference about the bioelectrical impedance analysis?
L 126-128. Is this definition of VO2 max supported by validation studies? Please add references.
L 150 please support this claim with a reference
L 184 Such precision on age (two digits) is not necessary
L 186 and Table 1. If you randomly selected a test set from the whole set, by definition, the train and test sets are similar. No need to present both sets separately, and no need to test for similarity: any significant results would be a type I error. See for instance https://www.ncbi.nlm.nih.gov/pmc/articles/PMC5947842/ for further information.
L 188-193. Were the results obtained from the test set? Please clarify and repeat here the sample size (N=?)
L 206 What were the results of the study of Nes et al.? If your aim is to compare your results with other study, please briefly comment on study’s outcome.

Table 2. Please indicate the sample size

Figures. Please add the sample size

---

## Round 0.2 · Minor Revisions

As outlined by the Reviewer below, please make the suggested changes to the tables and further improve English throughout the manuscript. For figures 4 and 5, please add in correlation coefficients and corresponding p values. Please also check that exercise was terminated due to bradypnea (i.e. slow breathing rate) as outlined in your methods section.

Reviewer 1 ·

Basic reporting

Q. Clear and unambiguous, professional English used throughout
A. No. This work requires improvement in English

Q.Literature references, sufficient field background/context provided.
A. Yes

Q. Professional article structure, figures, tables. Raw data shared
A.Yes

Q.Self-contained with relevant results to hypotheses.
A. Yes

Experimental design

Q. Original primary research within Aims and Scope of the journal.
A.Yes

Q.Research question well defined, relevant & meaningful. It is stated how research fills an identified knowledge gap.
A. Yes

Q.Rigorous investigation performed to a high technical & ethical standard.
A. Yes

Q.Methods described with sufficient detail & information to replicate.
A. Yes

Validity of the findings

Q. Impact and novelty not assessed. Negative/inconclusive results accepted. Meaningful replication encouraged where rationale & benefit to literature is clearly stated.
A. Yes

Q.All underlying data have been provided; they are robust, statistically sound, & controlled.
A. Yes

Q.Conclusions are well stated, linked to original research question & limited to supporting results.
A. Yes

Additional comments

Table 2 and Table 3. These tables should include all dataset, not just training set.

Table 4. Delete PRESS value. Press SEE and Press R2 is enough.

Also, please check English with native speaker of English and provide certificate if possible.

---

## Round 0.3 · accepted · Accept

Thank you for addressing the reviewer comments and undertaking professional English editing which has improved this manuscript.